# Molecular sensitised probe for amino acid recognition within peptide sequences

Xu Wu [1,2,9], Bogdana Borca [3,4,9], Suman Sen[1], Sebastian Koslowski[1], Sabine Abb[1], Daniel Pablo Rosenblatt[1], Aurelio Gallardo[5,6], Jesús I. Mendieta-Moreno [5], Matyas Nachtigall [5], Pavel Jelinek [5] ✉, Stephan Rauschenbach[1,7] ✉, Klaus Kern [1,8] & Uta Schlickum[1,3] ✉

The combination of low-temperature scanning tunnelling microscopy with a mass-selective electro-spray ion-beam deposition established the investigation of large biomolecules at nanometer and sub-nanometer scale. Due to complex architecture and conformational freedom, however, the chemical identification of building blocks of these biopolymers often relies on the presence of markers, extensive simulations, or is not possible at all. Here, we present a molecular probe-sensitisation approach addressing the identification of a specific amino acid within different peptides. A selective inter-molecular interaction between the sensitiser attached at the tip-apex and the target amino acid on the surface induces an enhanced tunnelling conductance of one specific spectral feature, which can be mapped in spectroscopic imaging. Density functional theory calculations suggest a mechanism that relies on conformational changes of the sensitiser that are accompanied by local charge redistributions in the tunnelling junction, which, in turn, lower the tunnelling barrier at that specific part of the peptide.

For the detailed understanding of biological functions, the atomically precise structural determination is essential. Scanning Probe Microscopy (SPM) operating at cryogenic temperatures is a versatile tool for single molecular level real-space investigation of structures and electronic properties with sub-nanometer resolution[1]. Recently, based on a combination with electrospray ion beam deposition (ESIBD)[2], SPM became available for the imaging of complex molecular structures[3–5] and biomolecules[6–8].

In ESIBD non-volatile molecules are transferred into the gas phase by electrospray ionisation[9,10] and are deposited intact on a clean surface in ultrahigh vacuum (UHV) after mass selection at a well-defined, low energy. In addition to probing the structure and properties of complex synthetic molecules[11–13], the combined ESIBD and STM

approach succeeded in imaging of biopolymers such as proteins[14,15], peptides[16,17], as well as saccharides and glycans[18–21]. The images reveal the structure at submolecular level showing building blocks of amino acids (AA) or monosaccharides and structural features like branches[19].

Using STM imaging to probe the sequence of biopolymers in peptides, proteins, or saccharides at the single molecule level would be a powerful tool in structural biology, helping to determine fundamental biological processes related to cellular activity, inter-cellular signalling, and chemical reactions for metabolism[22–30]. However, especially for flexible biopolymers, the difficulty in assignment of the observed SPM topography with an atomic scale model of the molecular structure grows with size and complexity of the investigated molecule. SPM imaging reveals a topography closely related to

[1]Max Planck Institute for Solid State Research, Stuttgart, Germany. [2]School of Integrated Circuits and Electronics, Beijing Institute of Technology, Beijing 100081, China. [3]Institute of Applied Physics and Laboratory for Emerging Nanometrology, Technische Universität Braunschweig, 38104 Braunschweig, Germany. [4]National Institute of Materials Physics, 077125 Magurele, Romania. [5]Institute of Physics of the Czech Academy of Science, Prague, Czech Republic. [6]Department of Condensed Matter Physics, Faculty of Mathematics and Physics, Charles University, Prague, Czech Republic. [7]Department of Chemistry, University of Oxford, Oxford, UK. [8]Institut de Physique, École Polytechnique Fédérale de Lausanne, Lausanne, Switzerland. [9]These authors contributed equally: Xu Wu, Bogdana Borca. ✉e-mail: jelinekp@fzu.cz; stephan.rauschenbach@chem.ox.ac.uk; u.schlickum@tu-bs.de

electronic density of states at the Fermi-level, which shows the shape of the molecule at sub-molecular level. While atomic resolution can be obtained from the substrate, for molecules the shape of the orbitals determines the observed topography[31–34]. The chemical information of the molecule is encoded with the electronic states, however it is hidden due to energetic and spatial overlap of molecular orbitals and substrate bands. Therefore, with the expectation of few special cases[35], tunnelling spectroscopy with a metallic tip rarely reveals chemically specific information directly.

Due to these limitations, the assignment of sequence or structure to a topography is only possible if supported by extensive theoretical investigations or obvious markers[18,36]. However, also simulation methods are limited to relatively simple model systems. With the growing size and complexity of the molecule the conformational space that would need to be considered to simulate SPM images for comparison, quickly becomes too large to handle[37].

While ESIBD deposits a chemically identified, mass-selected molecular ion, and the control over conformation is possible[15,20], the limitations in obtaining local chemical information can hamper the application of SPM to a single molecule structure determination of glycans[19] or peptides[16,17] for applications in a structural biology context, when topographic markers or modelling falls short. However, already partial, local chemical information on the molecular adsorbate could aid the sequence and structure assignment, because the global chemical information is given by the ESIBD process.

Recently, SPM imaging of organic molecules was greatly enhanced by the application of functionalised probes in non-contact AFM[8,38,39], where atomic resolution detail could be observed in the images of typically flat, polycyclic hydrocarbons[38,40,41]. This level of atomic detail can be sufficient for a full chemical identification[32,33]. However, the direct structural assignment to more complex, three-dimensional molecules applying high-resolution nc-AFM imaging with CO-modified probes is still difficult[33,42]. It has also been shown on small molecules that a modified probe functionalized with a specific molecule can give additional information from an analyte molecule via a

mechanism specific to the geometry of the modified probe[42–44]. For instance, sensitivity to amino acids and short peptides has also been demonstrated by non-imaging tunnelling methods through a gap sensitised with different sensitiser[45–48]. This suggests that probe modifications in SPM by molecules other than CO might reveal information beyond conventional STM or AFM imaging and spectroscopy, potentially useful towards chemical identification[32,34,49,50].

Following the idea of the probe functionalisation, here we present a molecular imaging strategy providing chemical contrast based on a selective interaction of a molecular sensitised STM tip with a moiety in a complex molecule on the surface. The sensitised probe is created by picking up a small, reactive sensitiser molecule derived from an amino acid (AA) pre-deposited to the surface. A such modified probe presents chemical sensitivity towards one specific amino acid, tryptophan, in a peptide composed of tryptophan (W), proline (P), and arginine (R). The sensitivity results as an enhanced feature in tunnelling spectra at the position of W. This feature is mapped via spectroscopic imaging to pinpoint spatially the location of W. We tested the approach within a set of different synthetic oligopeptides where we consistently found features at locations consistent with the position of the corresponding AA. The origin of the chemically specific interaction between the sensitised probe and W can be understood via a docking mechanism that alters the geometry as well as local charge distribution of the molecular probe and hence changes the tunnelling barrier, which is confirmed with density functional theory (DFT) calculations.

## Results and discussion
### Molecular sensitisation of the probe for amino-acid identification

We analyse several synthetic peptides containing different sequences of the three AA: W, P, and R (Fig. 1a) in sequences of length of eight to twelve AA. These three AA have distinctly different properties. W has a large flat aromatic group of a five and a six-atom ring with a slight polarity due to the N-atom in the five-ring, R has several charged

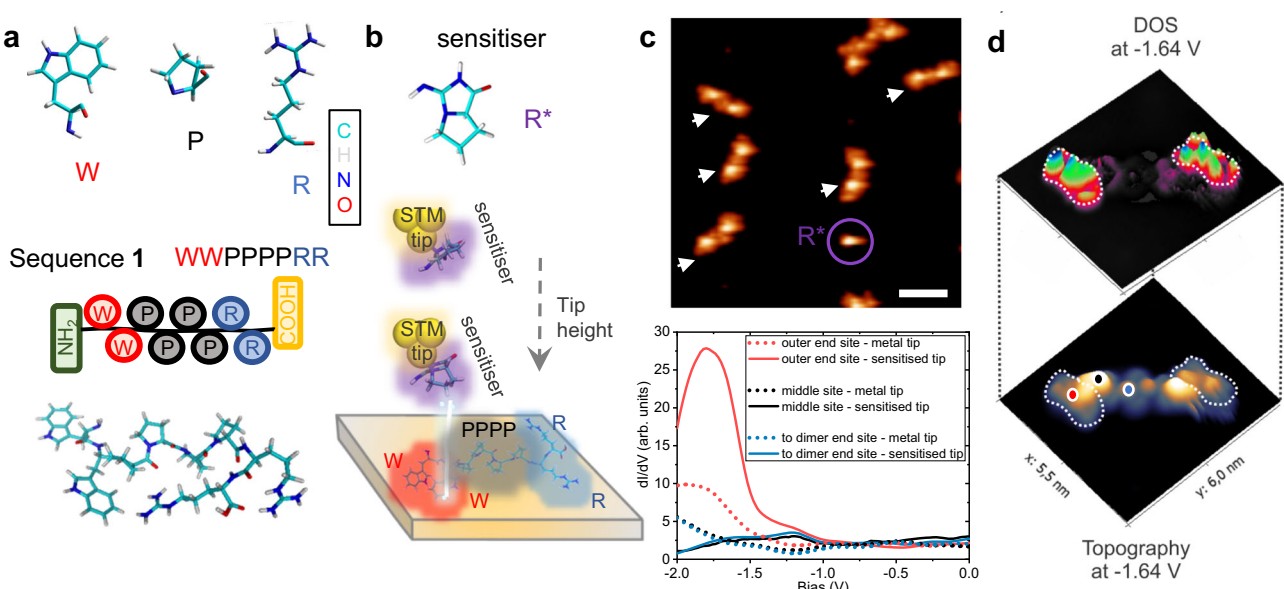

**Fig. 1 | Amino-acids identification by STM. a** Structural representation of the amino-acids sequence components, the tryptophan (W), the proline (P) and the arginine (R), and of sequence **1** (WW-PPPP-RR). **b** Sensitiser and the sensitisation procedure using a functionalised tip with the arginine fragment (R*), which is flexible at the tip apex, changes its conformation by approaching it towards the surface and enhances the tunnelling conductance features on specific parts of a peptide. **c** STM image of several dimers marked with arrows of the sequence **1**

formed on Au(111) where a sensitiser molecule R* is highlighted by a circle and spectroscopic traces acquired with a metal and a sensitised tip on different sites of a sequence as marked in panel (**d**) by points. **d** 3D illustration of a STM topographic image (bottom) and the tunnelling conductance map (d$I$/d$V$ map) (top) of a dimer. The d$I$/d$V$ map that is acquired with a sensitised tip at a characteristic bias voltage of −1.64 V, shows a significant feature enhancement towards the marked outer-end-site of each peptide in the dimer.

groups and no aromaticity, and P has an additional bond in the backbone, which leads to limited flexibility[15].

Generally, oligopeptides are not volatile, therefore we employ ES-IBD for the preparation of the sample, ensuring intact deposition by controlling the chemical composition and choosing a low deposition energy (Supplementary Results – Electrospray ion beam deposition and Methods section). The deposition of peptide **1**, sequence WW-PPPP-RR (Fig. 1a), on Au(111) leads to the formation of structures clearly identifiable as peptide dimers based on size, symmetry, and height (Fig. 1c). The brightest topographic feature at the centre of each monomer likely corresponds to the stiff proline PPPP section in the middle of the peptide[15,17]. The two tail sections of RR and WW, however, cannot be distinguished based on topography.

To determine the local position of one specific AA within the topographic image we sensitise the STM-tip apex with a fragment of an individual arginine molecule (R*) (Fig. 1b) deposited onto the surface by thermal sublimation and decomposition of arginine (Supplementary Results – Sensitiser). The R* fragment species is identified by chemical analysis to be a double ring creatinine-proline derivative ($C_6H_9N_3O$, m/z = 139)[51]. On the surface R* is imaged as one characteristic feature in topographic images (encircled in Fig. 1c). It is attached to the apex of the STM tip by a vertical manipulation procedure. With the goal of differentiating between the building blocks, we characterise the adsorbed peptides locally by tunnelling spectroscopy using metal tips as well as R*-sensitised probes. Differential conductance curves in the range of −2 V to 0 V are acquired at several positions across the molecule (Fig. 1c).

With an unmodified metal tip, a weak and broad shoulder appears in the tunnelling spectra below −1.5 V if the tip is placed at the outer edges of the peptide dimer (marked in Fig. 1d). Likewise, in the centre of each molecule and at the peptide terminal within the dimer centre weak spectral feature can be observed in this voltage region.

The sensitised tip significantly alters the spectroscopic characteristic at the outer edges, *i.e.* a pronounced peak appears around −1.7 V in the differential conductance (Fig. 1c). The d*I*/d*V* spectra measured in the centre of the peptide and at the peptide terminal within the dimer show reduced intensity at this voltage. Overall, spectra taken at a larger energy interval and at all other locations on the molecule remain featureless, suggesting they are nearly not affected by sensitisation in contrast to the metal-probe spectrum (Supplementary Results – Sensitiser).

Using DFT calculations, the sensitisation enhanced spectral feature observed below −1.5 V can be assigned to the highest occupied molecular orbital (HOMO) of the molecule, which is localised in the WW section of the peptide (Supplementary Results – Molecular orbital). The spatial distribution of this enhanced feature can be mapped by tunnelling conductance maps measured at the corresponding energy (Fig. 1d). These conductance maps show distinct, intense features at the far sides of the dimer, *i.e.* at each outer end of the single peptide, whereas the middle part of the molecule and the inner dimer interface are featureless. d*I*/d*V* maps recorded without tip functionalisation would show a large number of features of similar magnitude in which no one feature is distinct.

Consequently, the R*-sensitised tip reveals the location of the tryptophan motif (WW) within the peptide due to the strong contrast in the d*I*/d*V* maps recorded at −1.6 V. Then the position of the other AA of the peptide can be identified in accord with theoretical calculations that optimise the molecular arrangement of the peptides on the surface (Supplementary Results – Structural model).

## Enhancement mechanism of the sensitised probe

To rationalise the observation of the enhanced spectroscopic peak for the interaction of a specific AA with a sensitised probe, we performed quantum mechanics/molecular mechanics[52] (QM/MM) simulations of the tip-sensitiser-adsorbate system using the Fireball density

functional theory[53] (DFT), with the peptide in the QM region and the surface in the MM region. The simulations consider the conformation of the sensitised probe interacting with the substrate. First, we carried out an extensive search to find the optimal adsorption configuration of a single R* unit attached to an atomically sharp tip apex via its nitrogen atoms (Fig. 2a). In the minimum energy configuration found, the sensitiser was bound to the metallic tip apex by 1.9 eV. In addition, we found another configuration of a single R* unit with similar binding energy (Supplementary Fig. 6), while other configurations had much higher energies. However, according to our calculation, this configuration is not suitable for amino acid recognition, as we did not find a reduction of the tunnelling barrier upon interaction with the peptide on the substrate. In addition, we also carried out optimisation of the peptide on the surface including preliminary annealing to 300 K with subsequent optimisation (Supplementary Results – Structural model). The resulting structure was later employed for the simulations of tip-sample interaction.

Starting from that optimal geometry of the sensitiser molecule bound to the tip, the probe is approached towards the surface. At each step, the system is fully relaxed to adopt the optimal conformation at the current height given by the interaction between the sensitiser and the surface. Figures 2a, b, d, e display the evolution of tip and molecule conformation upon approach, revealing the flexibility of the sensitiser molecule at the tip apex. The probe flexibility facilitates the adaptation of an optimal configuration that accounts for the interaction with the target amino acid located beneath the tip.

According to the calculations, the conformation of the sensitised tip is mainly determined by the electrostatic and dispersion interaction with the adsorbate. This is observed in the comparison of the probe geometry when approaching to W or R section of a peptide (Fig. 2a, d). In both cases there is a change in the probe geometry. When the tip approaches a W residue a large modification in the tip-sensitiser geometry is observed. The R*-sensitiser rotates at the tip apex to create a strong cation-π interaction with the W molecule on the surface. In contrast, when the functionalised tip is approaching the R section of the peptide on the surface, only a small bend of the R*-sensitiser molecule at the tip apex is observed (Fig. 2d).

The origin of these changes in geometry can be understood by analysing the electrostatic potential surface (EPS) of the tip and of the peptide at two different tip heights for approaching a W and a R motif (Fig. 2b, e). In the surface adsorbed peptide, the tryptophan amino acid, W has different local charge distribution and polarisation than arginine, R. Thus, the sensitised probe is approaching to W with a different orientation of R* than to the R amino acid. In particular, the R amino acid is more polar and hence the tip-sensitiser prefers to approach with the negatively charged N atom to optimise the electrostatic interaction. In contrast, when the tip-sensitiser approaches the unpolar W residue it undergoes substantial rearrangement (Fig. 2a) to optimise dispersion interaction with the tryptophan amino acid W. This structural rearrangement of the sensitiser R*, causes a charge redistribution on W, which can be noted as a colour change in the EPS of the adsorbed W (Fig. 2b), while the EPS for an approach to R changes only slightly (Fig. 2e).

Importantly, the charge redistribution and the changes of geometry of the sensitiser are responsible for a local change of the tunnelling barrier. This can be observed as a difference in the Hartree potential when the tip approaches either W or R, see blue profile in Fig. 2c, f, respectively. A major reduction of the tunnelling barrier potential is observed for small tip-sample separations between R* and W amino acid, see Fig. 2c. This is only possible due to the relaxation of the R* sensitiser conformation due to site-specific interaction with particular amino acid. In the case of W amino acid, we observe depletion of electron density (forming local positive charge) which decreases the tunnelling barrier. On the contrary, in the case of R amino acid an accumulation of electron density is observed and

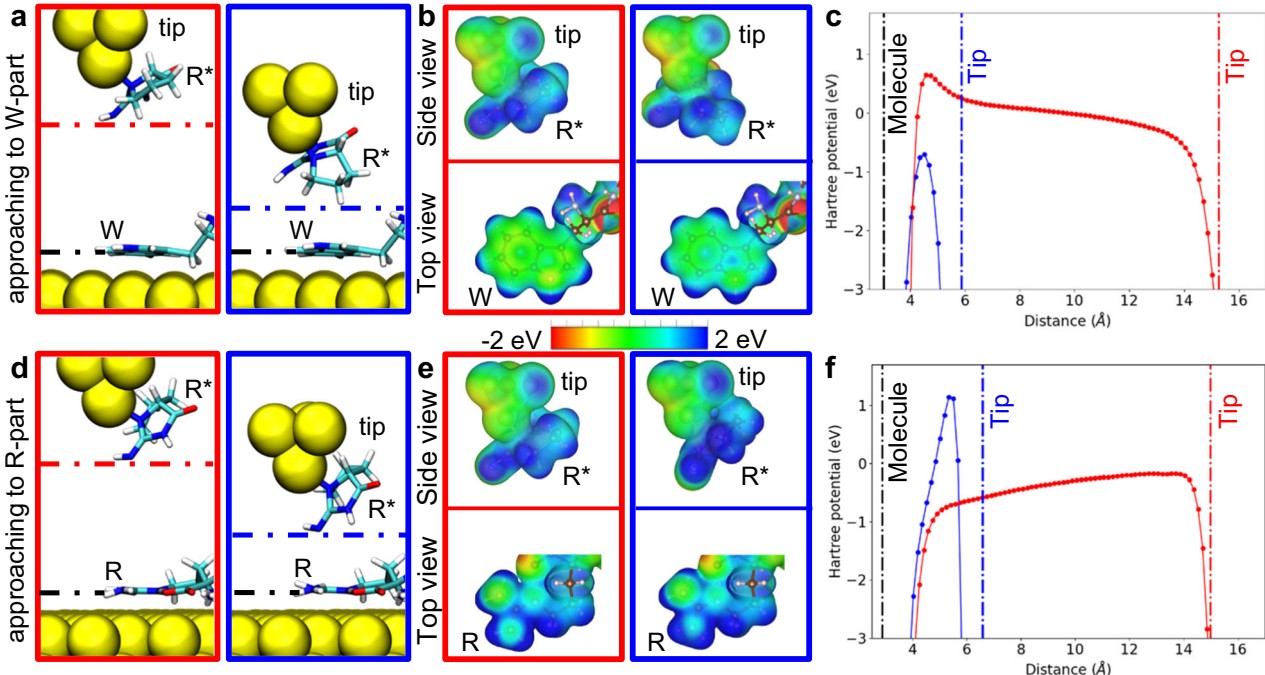

**Fig. 2 | Mechanism of the sensitisation. a, d** Conformational changes of the sensitiser at the tip apex when is approaching to the W (**a**) and to the R (**d**) part. The red and blue panels correspond to a tip-surface distance of about 15 Å and 6 Å, respectively (marked by vertical lines in the panels **c** and **f**). **b, e** Electrostatic potential surface (EPS) of the tip-sensitiser and of the W (**b**) and R (**e**) at the same tip heights as in panels (**a**) and (**d**). The potential over the 5-ring in W is notably reduced. **c, f** Hartree potential as a function of the tip height when approaches to the W (**c**) and to the R (**f**) section. The vertical lines represent the tip-surface distances in correspondence with the marked panels in **a, d** and **b, e**.

consequently the tunnelling barrier is enhanced, as shown in Fig. 2f. This different local variation of tunnelling barrier on specific AA leads to the enhancement of the feature in the d$I$/d$V$ conductance signal. The conformational freedom allows for the changes of the surface electrostatic potential and hence tunnelling barrier, consequently enhancing the differential tunnelling current at the position of the target AA of the peptide on the surface.

We can expect this interaction to be specific for the R*-W interaction, because it depends on a combination of several conditions which all need to be fulfilled: a site-specific interaction causing a conformational change on the sensitiser and local change of charge density, which modifies the tunnelling barrier. The specificity thus also depends on the conformation of the amino acid in the adsorbed peptide, which can be assumed to be similar for W, always being adsorbed flat on the metal surface. In general, this sensing concept can be extended to other combinations of functional groups, probe-sensitiser and amino acid in the peptide with a proper match of EPS patterns that gives rise to a site-specific interaction that causes a local change in conformation of the sensitiser as well as tunnelling barrier.

**Application to other sequences**
In order to correlate the appearance of an enhanced d$I$/d$V$ response with the interaction of one AA with a R*-sensitised probe, we studied peptides of different sequences with the presented approach. We prepared surfaces with several peptides of different sequences, all containing W, R, and P amino-acid motifs, however always at different locations in the peptide. For this we chose sequences which all form molecular dimers on the Au(111) surface, so that we are able to identify the single molecule and determine an arrangement of the peptide. Figure 3a, e, i, m shows the peptides **1, 2, 3** and **4** (WW-PPPP-RR, WW-PPP-RR-PPP-RR, WW-RR-PP-RR and W-PPP-W-PPP-RR) with the corresponding STM topography of a single dimer (Fig. 3b, f, j, n). The individual peptide can be identified by the symmetry of the dimer

structure and the orientation of the peptide within the dimer using our specific sensitised tip.

The corresponding tunnelling conductance maps taken at the characteristic energy of the enhanced spectral feature are shown for each sequence for one peptide of the dimer in Fig. 3c, g, k, o. Each of the d$I$/d$V$ maps of these other sequences (**2, 3, 4**) shows localised, enhanced intensity near the far side of the dimer, where the W motif is expected to be localised.

Even though the peptides have an elongated conformation and their long axis can be assumed to follow the backbone (Fig. 3b, f, j, n), the exact position of the AA cannot be simply inferred by the position of the observed lobes in the topography[14–18,33]. However, the close inspection of the observed patterns in the corresponding tunnelling conductance maps (Fig. 3c, g, k, o) compared to the motifs present in the sequences, confirms that the d$I$/d$V$ signal enhancement is related to the interaction of the sensitised tip with the W motifs of the peptides and allows for the assignment of the sequence (Fig. 3d, h, l, p).

To underpin our concept, we have structured the probed peptide sequences such that they create different environments for the sensitised probe-peptide interaction, while not changing the peptide itself too much. Sequence **1**: Amino acids of each type are placed at one specific position and W is neighbouring to P. Sequence **2**: Amino acids P and R are present twice at different locations; W is neighbouring to P. Sequence **3**: Amino acid R is located at two different locations and the place of the nearest neighbour amino acid is exchanged, here W is next to R. Sequence **4**: Amino acid W is located twice at two different positions

With this set of peptides, we can unambiguously proof to which amino acid our R* tip is sensitive. A first key observation is that peptide **2** (WW-PPP-RR-PPP-RR) only shows one location of enhanced d$I$/d$V$ intensity placed at one end of the peptide sequence. This position can only be ascribed to the WW motif since RR and PP motifs are each present twice within this sequence and are spatially well separated. Next, peptide **3** (WW-RR-PP-RR) does

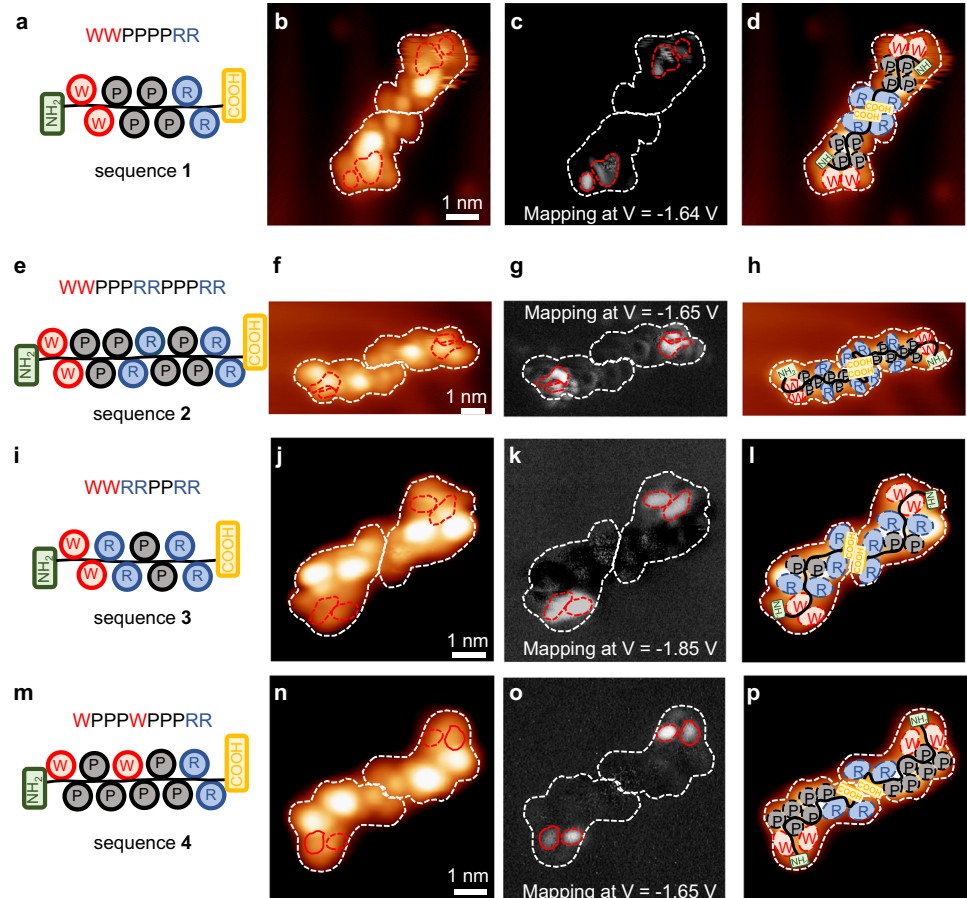

**Fig. 3 | Application to several sequences. a**, **e**, **i**, **m** Representation of the **1** (WW-PPPP-RR), **2** (WW-PPP-RR-PPP-RR), **3** (WW-RR-PP-RR), and **4** (W-PPP-W-PPP-RR) peptides. **b**, **f**, **j**, **n** STM topographic images of dimers formed by the corresponding peptides. **c**, **g**, **k**, **o** Enhanced tunnelling conductance maps of the respective peptides. **d**, **h**, **l**, **p** Proposed models of the peptide sequences identified on the STM images.

not contain a WP motif, but only contains the WR mixed motif in the sequence. Likewise, sequence **4** does not contain a WR motif, and hence both, WR and WP can be excluded as origin of the enhancement. Therefore, that observation excludes that the d$I$/d$V$ feature resulting from a mixed motif of WR or WP. It only correlates to the enhancement with the W amino-acid features. In addition, the tunnelling conductance map of sequence **4** (W-PPP-W-PPP-RR) (Fig. 3m) shows two slightly further separated features, which are consistent with locations of the two W in the peptide which are separated by a PPP section. The investigation of these different sequences confirms the sensitivity of the specific functionalised tip to the amino acid W. Since we only observe the enhancement at the local position of W, our method can be transferred to any other sequences being able to locally identify W.

In conclusion, our results demonstrate the identification of the spatial position of the tryptophan amino acid W within peptide sequences. We observed an enhanced d$I$/d$V$ signal at the local position of W using a R* sensitised probe that is generated by a reduction of the barrier and a concurrent conformational change at the sensitised probe. This enhancement is a specific occurrence and allows the allocation of W within a complex molecule like the peptides of 8–12 amino acids as used in our study and is thus a first step towards sequencing of peptides at surfaces. A d$I$/d$V$ image, in which many features in the same magnitude range would be visible without sensitisation, now presents singular features of much larger contrast. Because the structural change at the sensitised probe enhances some features while suppressing others, the discovered effect allows for chemically specific imaging.

Molecular modification of SPM probes has already been extremely successful in many different cases, most importantly enhancing the resolution of STM and AFM images. Our work shows that specific sensitisation at the tip apex is a much more general route towards extracting additional information from a complex molecular system. It is also worth to note that the concept can be in principle extended to atomic force microscopy, where the site-specific interaction between the sensitiser and amino acid will be encoded in recorded frequency shift signal.

Generally, a sensitiser molecule attached to the apex of the STM tip has to fullfil several criteria to promote chemically specific, high-resolution imaging. (i) Once bound to the probe, the sensitiser molecule should be chemically inert, avoiding covalent bond with inspected molecules; (ii) it should be flexible to enhance the detecting signal via the proposed mechanical transducer mechanism, and (iii) enhanced polarity is also desired as it makes the probe more sensible to spatial variation of charge distribution on samples; (iv) a molecular configuration beyond a linear configuration such as CO-tip (often used for tip functionalisation)[33,38,39,54,55], may impose more complex docking mechanism, which can make the probe more specific to single units within complex molecular pattern such as peptides, proteins or glycans.

The information obtained from specific interactions with a such sensitised probe offers additional information which can be instrumental to solve the structures of complex molecules[56,57] at the sequence/subunit level, reducing the extent of calculations needed or making them feasible to begin with. It enables the effective analysis of larger peptide structures, even entire unfolded proteins and their

assembly at surfaces, which would have otherwise impeded by the complexity of the system in light of the limited chemical information accessible in conventional SPM topography. Likewise, an unknown natural sequence or polymers and complex molecules deposited on surfaces without mass-selection[58–61] could be partly or entirely identified, relying on our sensitisation procedure in combination with a database amino-acid sequences in proteins[62,63]. In combination with ES-IBD for the preparation of single molecules at surfaces with highest quality, the low temperature scanning tunnelling microscope with molecular-sensitised probes becomes an excellent tool for individual single molecule biopolymer analysis, providing highest structural resolution and local chemical information.

## Methods

### Samples preparation

Several synthetic peptides with the desired sequence were deposited onto Au(111) surface using the mass-selective ESIBD technique. For electrospray ionisation the peptide solution was diluted with water and ethanol and approx. 1% formic acid was added. Time-of-flight mass spectra were measured of the initial ion beam and the mass-filtering rf-quadrupole was adjusted to select only one ion species for deposition (Supplementary Fig. 9).

In addition to peptides deposited by ES-IBD, a small sensitiser molecule is also deposited, but through a thermal sublimation procedure, on the same surface and can be clearly recognised by its smaller size and then picked up at the tip apex by a vertical manipulation technique in STM.

Before the peptide deposition, the atomically flat Au(111) single crystal (procured from Surface Preparation Laboratory, SPL) was cleaned in a UHV chamber by repeated cycles of Ar$^+$ ion bombardement (with an energy of 1 keV and a partial pressure of $1 \times 10^{-6}$ mbar for 20 min) and annealing (at 800 K for 10 min). After the cleaning, the substrate was transferred via an UHV suitcase (with a base pressure of $2 \times 10^{-10}$ mbar) to the UHV chamber of the ES-IBD device. After the peptide deposition, the sample was transferred back into the preparation chamber and the sensitiser R* molecules were deposited onto the surface by thermal evaporation (at 185 °C for 30 s) leading to a sub-monolayer coverage (details in the SI). L-arginine (R), was bought as an enantiomer pure powder from Sigma Aldrich (Purity > 99%). Prior to the R deposition the sample was cooled at about 77 K.

### STM measurements

After each preparation, the sample was transferred under UHV conditions (not more than $8 \times 10^{-10}$ mbar for the whole preparation and transfer procedure) to the LT-STM (operated at 4.5 K). The LT-STM is a home-built system controlled by Nanonis RC5 electronics. The STM experiments were performed with a chemically etched W tip. The tip functionalisation was realised by placing the STM tip on top of the desired sensitiser molecule that was imaged beforehand, then the feedback loop is switched off at a bias voltage of −1 V and the tip is approached towards the sample by 200–300 pm. At the favourable distance of the tip above the sensitiser where the attractive forces are dominant, the sensitiser is attached to the tip. After that, the same area is scanned again to make sure that the R* fragment is at the tip apex and check the tip state on the Au(111) surface with d$I$/d$V$ curves (Supplementary Results – Sensitiser). When needed, the sensitiser molecules are removed from the STM tip by applying high voltages and field emission procedures. The tunnelling conductance measurements (d$I$/d$V$ spectra and maps) were carried out by using a standard lock-in technique with a modulation frequency of 832 Hz and amplitude of 50 mV. The enhanced tunnelling conductance maps were acquired with a sensitised tip at the defined bias voltage in constant current mode. In all experiments, the bias voltage was applied to the sample. The WSxM software[64] was employed for STM data analysis.

### DFT calculations

We employed QM/MM (quantum mechanics/molecular mechanics) methodology with the DFT package Fireball[52]. The QM Fireball[53] calculations use the BLYP[65,66] exchange correlation functional with D3 corrections[67] and norm-conserving pseudopotentials, together with a basis set of optimised numerical atomic-like orbitals[68] with a s orbital for H and sp$^3$ orbitals for C,N,O. The MM part uses the interface force-field[69] for the surface and parm99sb for the peptide[70]. The whole peptide and the tip- sensitiser has been treated with the QM formalism and the 3 layer Au[111] surface with the MM force field. A model of metallic tip apex was made of 4 atoms adopting a pyramide-like configuration on Au[111] base. To find the final structure we started with a classical MD annealing to 100 K followed by QM/MM final relaxation. The interaction between the functionalised tip and the peptide was simulated, fixing the position of the metallic part of the tip-sensitiser at different heights with respect to the peptide and performing a short QM/MM MD (1 ps) at 100 K for each height. We started the scan at a sensitiser-peptide distance of 12 Å and approached with steps of 0.2 Å up to a distance of 2 Å. This simulation was followed by a geometry relaxation at each tip-sample distance. At each point, profile of the Hartree potential and Electrostatic Potential Surface were calculated for the fully optimised atomic structure. For the tip approach calculations, the whole tip was treated with the quantum treatment. The energy levels and molecular orbitals were calculated in gas phase using the code Fireball, with the coordinates of the peptide absorbed on the surface and relaxed with QM/MM.

## Data availability

All data underlying this study are available from the corresponding authors upon request.

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

## Acknowledgements

We acknowledge funding by the Deutsche Forschungsgemeinschaft (DFG, German Research Foundation) under Germany's Excellence Strategy–EXC-2123 QuantumFrontiers– 390837967, We acknowledge funding by the Emmy-Noether-Program of the Deutsche Forschungsgemeinschaft, B.B. acknowledge the Romanian Ministry of Research, Innovation and Digitalization for funding through UEFISCDI of the project PN-III-P2-2.1-PED-2021-0378 (contract nr. 575PED⁄2022) and the Core Program PC2-PN2308020. A.G., J.M., M.N., and P.J. acknowledge financial support from the CzechNanoLab Research Infrastructure supported by MEYS CR (LM2023051), the GACR project no. 20-13692X and computational resources were provided by the e-INFRA CZ project (ID: 90254), supported by MEYS CR.

## Author contributions

U.S., S.R. and K.K. conceived this project and coordinated the experiments. X.W., S.S., S.K. S.A., S.R. and D.R. carried out sample preparations and performed the experiments. X.W., B.B. and S.K. analysed experimental data and supported the conceptual design of model systems. A.G., J.I.M.-M. and and M.N. performed the theoretical calculations under the guidance and support of P.J. All authors participated in discussions that helped guide various aspects of this manuscript. The first draft of the manuscript was written by B.B., S.R. and U.S. All authors contributed to the editing of the manuscript.

## Funding

## Competing interests

The authors declare no competing interests.
