## [Peer Review File · Nature Communications]

REVIEWER COMMENTS

Reviewer #1 (Remarks to the Author):

The manuscript "Molecular sensitised probe for amino acid recognition within peptide sequences" by Wu et al. describes an approach to characterising peptide sequences using scanning tunnelling microscopy (STM). Peptides are deposited onto a Au(111) substrate via mass-selective electrospray ion beam deposition (ESIBD), and subsequent functionalisation of the STM probe using a sensitizer molecule (a molecular fragment, R*, created via thermal deposition) allows identification of one specific amino acid within several peptides.

The use of a molecular-functionalised STM probe to identify the location of specific amino acids has significant appeal and is an important progression of the capabilities of SPM as a characterisation technique. As such I think the work is appropriate for publication in Nature Communications.

However, there are several points that the authors should address before a revised version of the manuscript would be publishable.

1) The exact nature of the sensitizer species (and its adsorption at the tip apex) is an important point. The theoretical calculations show a sensible rationale for deducing a likely adsorption geometry. However, the tip apex of the 'real' STM probe is often poorly defined and one could imagine various ways the sensitizer molecule could attach to a less ordered tip apex. The authors should discuss this in the manuscript. Do they believe that more than one adsorption geometry is possible? Would it be possible to utilise the carbon-monoxide front atom identification (COFI) approach to gain more information about the nature of sensitizer-tip adsorption site? It would also be good to state how many experimental attempts were made at tip functionalisation and whether they all resulted in similar topographic images (which could imply that the sensitizer has adsorbed in a similar geometry in all cases).

2) Following on from the discussion of the molecule 'fragment', R*, used to sensitise the tip: The structure/nature of the species on the surface following thermal deposition does not seem to be well characterised by STM. How do the authors confirm that the species they use to functionalise the tip is indeed R*? This is obviously important as it underpins the computational work presented. A robust characterisation of the nature of this species should be provided. The authors provide a reference to the decomposition of arginine (Ref 50 in the manuscript) but the reference seems to imply that different tautomeric forms of the decomposed R* are possible. The authors should comment on why the calculations discussed in the manuscript chose the specific fragment. Were other tautomers considered? [The relevant text from reference 50 is – "This final residue is remarkable. It contains the proline ring, the guanidine star and a peptide bond in the ring of creatinine, which is the 5-ring with the =O and =OH double bonds. Creatinine, $H_f = -240$ kJ/mol, m.p. 300 °C, C₄H₇N₃O, chemspider 568, has several tautomeric forms. The end product in question might contain either of those rings. We have no way to decide between the alternatives, but a double ring structure seems likely."]

3) It would be good to see a comparison between the dI/dV maps taken with the sensitised tips and those with a metal (non-functionalised) tip. At present it is hard to judge the benefit of the sensitised tip over a metal one – the dI/dV spectra are clearly different but it's not clear how much different the tip functionalisation would make for the dI/dV maps. It would be good if the authors could include data showing dI/dV maps for a metallic tip.

4) The authors provide a nice background to the use of ESIBD in combination with STM. However, there seems to be only limited to reference to the 'non mass selective' variant – which has also yielded results (examples of recent publications include: *Nanoscale*, 2018,10, 1337-1344, *Chem. Commun.*, 2017,53, 1168-1171, *Phys. Rev. Lett.* 125, 206803, *ACS Materials Lett.* 2021, 3, 10, 1503–1512). As there is a wealth of publications in this area, citations to prior work should be expanded upon.

Reviewer #2 (Remarks to the Author):

This manuscript describes experimental detections of individual amino acids within short synthetic peptides using scanning tunneling microscopy aided by a sensitizer molecule. In contrast to transitional STM imaging of the peptides, the use of a sensitizer molecule enhances contrast of the STM spectra from individual amino acids, making it possible to accurately distinguish one particular type (tryptophan) from other two (proline and arginine). The experimental outcomes are rationalized through quantum mechanics / molecular mechanics simulations that find the enhanced spectral contrast to originate from rearrangements of the sensitizer molecules when approaching the target amino acid. Following that, the authors image several variants of peptides composed of the same three amino acids, showing how peptide identification may work in principle.

Overall, the manuscript describes an interesting study that demonstrates the use of recognition interactions for identification of select amino acids vi conventional STM setup. Using molecular simulations to determine the mechanism enabling experimental differentiation of amino acids is a particularly strong aspect of the work. The manuscript, however, falls short on demonstrating a tangible path toward protein sequencing, if that was the original motivation for the study.

Novelty: The principle of recognition tuning has been around for a while and has already been used for identification of amino acids, for example, by the Lindsay lab (ASU) using break junctions. The novelty of the current work lays in adopting the principle to a more conventional STM setup where a molecule is adsorbed on the surface.

Generality: It is not at all clear that the principle described can be extended to peptides that are not composed of the three amino acids used in this study. Have the authors attempted to image other peptides? What were the results?

Rigor: The protein sequence assignment shown in Figure 3 looks very impressive, at first. However, the images show *proposed* assignments, which have not been independently validated. The study would benefit from a more systematic investigation of peptide detection capability using, for example, a mixture of peptides of known and unknown sequences, dilution experiments, blind test, etc. If the goal of the study is finding a path to protein sequencing, such kind of data should be presented or at the very least discussed.

References: The literature review is outdated. The field of protein sequencing has moved on by a lot since 2018, please refresh your description of the field.

Reply to the referees:

Reviewer 1.

“The manuscript “Molecular sensitised probe for amino acid recognition within peptide sequences” by Wu et al. describes an approach to characterising peptide sequences using scanning tunnelling microscopy (STM). Peptides are deposited onto a Au(111) substrate via mass-selective electrospray ion beam deposition (ESIBD), and subsequent functionalisation of the STM probe using a sensitizer molecule (a molecular fragment, R, created via thermal deposition) allows identification of one specific amino acid within several peptides.*

The use of a molecular-functionalised STM probe to identify the location of specific amino acids has significant appeal and is an important progression of the capabilities of SPM as a characterisation technique. As such I think the work is appropriate for publication in Nature Communications.

However, there are several points that the authors should address before a revised version of the manuscript would be publishable.

Response:

We acknowledge the reviewer for the constructive comments and criticism thanks to which we have revised and improved our manuscript.

“1) The exact nature of the sensitizer species (and its adsorption at the tip apex) is an important point. The theoretical calculations show a sensible rationale for deducing a likely adsorption geometry. However, the tip apex of the ‘real’ STM probe is often poorly defined and one could imagine various ways the sensitizer molecule could attach to a less ordered tip apex. The authors should discuss this in the manuscript. Do they believe that more than one adsorption geometry is possible? Would it be possible to utilise the carbon-monoxide front atom identification (COFI) approach to gain more information about the nature of sensitizer-tip adsorption site? It would also be good to state how many experimental attempts were made at tip functionalisation and whether they all resulted in similar topographic images (which could imply that the sensitizer has adsorbed in a similar geometry in all cases).”

Response:

We agree with the referee that the sensitizer molecules may adsorb on different conformations at the apex of the STM tip, and it represents a fundamental aspect of sensitisation. We started our calculations by first performing an extensive search for the most stable conformation of the sensitizer molecule at the tip apex. According to our DFT calculations, we found two dominant configurations with significantly lower binding energy than the other configurations. These two configurations, shown in the image of R1, have very similar energies (~0.03 eV). However, according to the DFT calculation, the second configuration of the sensitizer molecule does not provide adequate sensibility for recognition. This finding is consistent with experimental observations, where only a part of picked-up sensitizer molecules was adequate for sensitisation. While accurate statistics were not recorded, it usually takes several (less than 10) attempts to find a suitable tip.

We modified the main manuscript accordingly: “First, we carried out an extensive search to find the optimal adsorption configuration of a single R* unit attached to an atomically sharp tip apex via its nitrogen atoms (see Fig 2a). In the minimum energy configuration found, the sensitizer was bound to the metallic tip apex by 1.9 eV. In addition, we found another

configuration of a single R* unit with similar binding energy (see SI-Fig S6), while other configurations had much higher energies. However, according to our calculation, this configuration is not suitable for amino acid recognition, as we did not find a reduction of the tunnelling barrier upon interaction with the peptide on the substrate.”

In addition, in the revised manuscript we have included the supplementary figure (Figure S6), which displays two dominant configurations.

Fig. R1. Relaxed structure, EPS and total energy of the two considered configurations. A) The structure presented in the main text in figure 2, presents a polar character. B) An alternative configuration with a similar energy that does not present the necessary polar character to achieve electrostatic selectivity.

While the CO-terminated tip was found very useful for the chemical resolution of planar molecules, we don't think that it can be used for the recognition of amino acids. Namely, its relatively simple structure and weak electrostatic quadrupole moment hinder the docking mechanism required for recognition.

Experimentally, we have developed an efficient strategy to reproducibly functionalise the tip. We select the sensitizer molecules in the STM images firstly based on their size and shape. Then, the molecules with characteristic features in the STS dI/dV curves, are chosen to be attached to the tip apex. Furthermore, the functionalized tips that are used for sensitisation show a distinctive STS curve acquired on the clean Au(111) surface. The characteristic STS features of the sensitizer on the surface and at the tip apex and the functionalization procedure are already presented in the supplementary file, Figure S3, and the corresponding description of it.

“2) Following on from the discussion of the molecule ‘fragment’, R*, used to sensitise the tip: The structure/nature of the species on the surface following thermal deposition does not seem to be well characterised by STM. How do the authors confirm that the species they use to functionalise the tip is indeed R*? This is obviously important as it underpins the computational

work presented. A robust characterisation of the nature of this species should be provided. The authors provide a reference to the decomposition of arginine (Ref 50 in the manuscript) but the reference seems to imply that different tautomeric forms of the decomposed R^* are possible. The authors should comment on why the calculations discussed in the manuscript chose the specific fragment. Were other tautomers considered? [The relevant text from reference 50 is – “This final residue is remarkable. It contains the proline ring, the guanidine star and a peptide bond in the ring of creatinine, which is the 5-ring with the =O and =OH double bonds. Creatinine, $H_f = -240$ kJ/mol, m.p. 300 °C, $C_4H_7N_3O$, chemspider 568, has several tautomeric forms. The end product in question might contain either of those rings. We have no way to decide between the alternatives, but a double ring structure seems likely.”

Response:

We agree with the referee that it is very important to identify the nature of the sensitizer molecule. Firstly, we have analysed this with two independent external mass spectrometers. Representative spectra have been shown and discussed in the supplementary file, Figure S2. The product molecule has the same mass as the molecules presented in reference 50. Furthermore, for sensitisation, the molecules are analysed based on the STS features following the procedure described above. Bibliographic literature (for instance, current Ref. 53) proposes the most likely product molecule resulting by Arginine sublimations/decomposition. There are two suggested tautomers from the respective chemspider search. Our supplementary calculations (added in the section Sensitizer of the SI file-Figure S5) show that after the attachment of R^* unit, the other possible tautomer is energetically much higher in energy (~ 0.73 eV). Therefore, we can conclude that the tautomerization process is quenched after the tip functionalization.

Fig. R2. Relaxed structure, EPS and total energy of the two considered configurations. A) The structure presented in the main text in Figure 2, that presents a polar character. B) Tautomeric configuration for comparison.

“3) It would be good to see a comparison between the dI/dV maps taken with the sensitised tips and those with a metal (non-functionalised) tip. At present it is hard to judge the benefit of the

sensitised tip over a metal one – the dI/dV spectra are clearly different but it’s not clear how much different the tip functionalisation would make for the dI/dV maps. It would be good if the authors could include data showing dI/dV maps for a metallic tip.”

Response:

The best way to quantitatively identify the properties of a tip apex is to compare dI/dV spectra at specific positions since this provides information about a wide range around the Fermi energy. Thus, taking spectra for different tip-apex-structures at the same position allows a direct comparison of the electronic properties for the different environments, as shown in Fig. 1d. In contrast, a dI/dV map only provides insights at one specific voltage. Therefore, a map taken with a metallic tip would not give any further information, showing relatively even intensity all across the molecule of similar appearance as the normal STM image.

“4) The authors provide a nice background to the use of ESIBD in combination with STM. However, there seems to be only limited to reference to the ‘non mass selective’ variant – which has also yielded results (examples of recent publications include: Nanoscale, 2018,10, 1337-1344, Chem. Commun., 2017,53, 1168-1171, Phys. Rev. Lett. 125, 206803, ACS Materials Lett. 2021, 3, 10, 1503–1512). As there is a wealth of publications in this area, citations to prior work should be expanded upon.”

Response:

We thank the referee for the suggestion, we have added the recommended references (new Ref. 60-63) in the outlook sentence that reads: “Likewise, an unknown natural sequence or polymers and complex molecules deposited on surfaces without mass-selection⁶⁰⁻⁶³ could be partly or entirely identified, relying on our sensitisation procedure in combination with a database amino-acid sequences in proteins.^{64-65”}

Reviewer 2.

“This manuscript describes experimental detections of individual amino acids within short synthetic peptides using scanning tunneling microscopy aided by a sensitizer molecule. In contrast to transitional STM imaging of the peptides, the use of a sensitizer molecule enhances contrast of the STM spectra from individual amino acids, making it possible to accurately distinguish one particular type (tryptophan) from other two (proline and arginine). The experimental outcomes are rationalized through quantum mechanics / molecular mechanics simulations that find the enhanced spectral contrast to originate from rearrangements of the sensitizer molecules when approaching the target amino acid. Following that, the authors image several variants of peptides composed of the same three amino acids, showing how peptide identification may work in principle.

Overall, the manuscript describes an interesting study that demonstrates the use of recognition interactions for identification of select amino acids vi conventional STM setup. Using molecular simulations to determine the mechanism enabling experimental differentiation of amino acids is a particularly strong aspect of the work. The manuscript, however, falls short on demonstrating a tangible path toward protein sequencing, if that was the original motivation for the study.

Novelty: The principle of recognition tunneling has been around for a while and has already been used for identification of amino acids, for example, by the Lindsay lab (ASU) using break junctions. The novelty of the current work lays in adopting the principle to a more conventional STM setup where a molecule is adsorbed on the surface.”

Response:

We thank the reviewer for the constructive criticism of our work, based on which we have revised and improved our manuscript. We hope that the reviewer will consider the revised manuscript suitable for publication in Nature Communications.

The manuscript is focused on the sensitisation procedure as an advancement of the competences of the SPM techniques. We have applied the method to identify a specific amino acid within predefined individual single peptide sequences. The molecular functionalization, mostly with CO molecules, of the SPM tips has successfully been used already for enhancing the spatial resolution of STM and AFM images, in many different cases, especially of π systems and flat molecules adsorbed on surfaces. Our work goes a significant step further, we show that a specific sensitisation with a more complex molecule at the tip apex can provide additional information, here chemical recognition through selective interactions. Thus, our new method leads to highest structural resolution and local chemical information.

“Generality: It is not at all clear that the principle described can be extended to peptides that are not composed of the three amino acids used in this study. Have the authors attempted to image other peptides? What were the results?”

Response:

In our paper, we show that we invented a new probe that is only sensitive to one specific amino acid, namely tryptophan, in a complex environment made of three AA with additional conformational freedom. To evidence this sensitivity and specificity, thus the chemical recognition of tryptophan, we probed several peptide sequences, all of them containing

tryptophan at different positions along the sequence. Based on these sequences, we can test that our sensitized probe is indeed only sensitive to the amino acid tryptophan.

It is correct that all peptide sequences contain only three amino acids, tryptophan, proline and arginine. The difference in the sequence leads to responses which allows for the conclusion that the observed enhancement is specific to W, while we maintain an environment, which is sufficiently similar between the sequences that our assumptions are maintained (*i.e.* peptide dimer, mostly linear, interacting via R residues at the C-terminal).

We understand the point of the referee that this does not represent a general peptide-sequencing scheme in itself. However, as the aim of the study is the establishment of a novel, chemically sensitive tip functionalisation in SPM spectroscopic imaging we feel that here we provide a reasonable proof of principle.

To underpin that the proof-of-principle is indeed plausible by our choice of peptide sequences, we rewrote the following paragraph:

“To underpin our concept, we have structured the probed peptide sequences such that they create different environments for the sensitised probe-peptide interaction, while not changing the peptide itself too much. Sequence 1: Amino acids of each type are placed at one specific position and **W** is neighbouring to **P**. Sequence 2: Amino acids **P** and **R** are present twice at different locations; **W** is neighbouring **P**. Sequence 3: Amino acid **R** is located at two different locations and the place of the nearest neighbour amino acid is exchanged, here **W** is next to **R**. Sequence 4: Amino acid **W** is located twice at two different positions.

With this set of peptides, we can unambiguously proof to which amino acid our **R*** tip is sensitive to the interaction with W. A first key observation is that peptide 2 (WW-PPP-RR-PPP-RR) only shows one location of enhanced dI/dV intensity placed at one end of the peptide sequence. This position can only be ascribed to the **WW** motif since **RR** and **PP** motifs are each present twice within this sequence and are spatially well separated. Next, peptide 3 (WW-RR-PP-RR) does not contain a **WP** motif, but only contains the **WR** mixed motif in the sequence. Likewise, sequence 4 does not contain a **WR** motif, and hence both, **WR** and **WP** can be excluded as origin of the enhancement. Therefore, that observation excludes that the dI/dV feature resulting from a mixed motif of **WR** or **WP**. Hence it correlates the enhancement with the **W** amino-acid features only. In addition, the tunnelling conductance map of sequence 4 (W-PPP-W-PPP-RR) (**Fig. 3m**) shows two slightly further separated features, which are consistent with locations of the two **W** in the peptide which are separated by a **PPP** section. Hence, the investigation of these different sequences confirms the sensitivity of the specific functionalised tip to the amino acid **W**. Since we only observe the enhancement at the local position of **W**, our method can be transferred to any other sequences being able to locally identify **W**.”

*“Rigor: The protein sequence assignment shown in Figure 3 looks very impressive, at first. However, the images show *proposed* assignments, which have not been independently validated. The study would benefit from a more systematic investigation of peptide detection capability using, for example, a mixture of peptides of known and unknown sequences, dilution experiments, blind test, etc. If the goal of the study is finding a path to protein sequencing, such kind of data should be presented or at the very least discussed”*

Response:

Because we have used mass-selective electrospray ion beam deposition techniques, the chemical structure of the studied molecules is precisely known. Hence, the assignment of the sequence has very little uncertainty.

Moreover, the proposed models of the molecular structure are based on theoretical calculations (presented for sequence 1 in the supplementary information in the Structural Model section). The dimer model of other sequences fits perfectly with the image and another arrangement is difficult to imagine.

“References: The literature review is outdated. The field of protein sequencing has moved on by a lot since 2018, please refresh your description of the field.”

Newer references from 2022 and 2023 (current Ref. 23-25) were added in the revised manuscript to the phrase that reads: “Using STM imaging to probe the sequence of biopolymers in peptides, proteins, or saccharides at the single molecule level would be a powerful tool in structural biology, helping to determine fundamental biological processes related to cellular activity, inter-cellular signalling, and chemical reactions for metabolism.²²⁻³⁰”

REVIEWERS' COMMENTS

Reviewer #1 (Remarks to the Author):

The authors revised version of the manuscript ("Molecular sensitised probe for amino acid recognition within peptide sequences"), SI, and responses address the majority of the points raised in the original review.

The additional detail on the possible adsorption geometries of R* at the tip apex are very helpful, as are the comments on the tautomers of the R* fragment.

The only point which is not really addressed is a discussion of how much sensitising the metal tip with the R* species effects the characterisation of the peptide sequence. Figure S4 presents dI/dV spectra acquired at different points along sequence 1. From these spectra it is clear that there is a resonance at around -1.7V for the functionalised and metal tips when positioned above the end of sequence 1, but not at other locations. On this basis, a dI/dV map taken at -1.7V using a metallic tip should show contrast at the end of the sequence, but not at other locations. Therefore, one should be able to determine the location of W amino acids without the aid of a sensitised tip. It is obvious that the sensitised tip should lead to a greater contrast above the W amino acid within the dI/dV map. However, the question of whether the sensitised tip is essential for amino acid recognition, or whether it enhances a signature which is still visible with a non-sensitised metallic tip, is important and should be explicitly commented on within the manuscript.

Following inclusion of this additional information, I would recommend the manuscript for publication in Nature Communications.

Reviewer #2 (Remarks to the Author):

The authors have addressed all comments raised in the previous round of review

Reviewer #1 (Remarks to the Author):

The authors revised version of the manuscript (“Molecular sensitised probe for amino acid recognition within peptide sequences”), SI, and responses address the majority of the points raised in the original review.

The additional detail on the possible adsorption geometries of R at the tip apex are very helpful, as are the comments on the tautomers of the R* fragment.*

The only point which is not really addressed is a discussion of how much sensitising the metal tip with the R species effects the characterisation of the peptide sequence. Figure S4 presents dI/dV spectra acquired at different points along sequence 1. From these spectra it is clear that there is a resonance at around $-1.7V$ for the functionalised and metal tips when positioned above the end of sequence 1, but not at other locations. On this basis, a dI/dV map taken at $-1.7V$ using a metallic tip should show contrast at the end of the sequence, but not at other locations. Therefore, one should be able to determine the location of W amino acids without the aid of a sensitised tip. It is obvious that the sensitised tip should lead to a greater contrast above the W amino acid within the dI/dV map. However, the question of whether the sensitised tip is essential for amino acid recognition, or whether it enhances a signature which is still visible with a non-sensitised metallic tip, is important and should be explicitly commented on within the manuscript.*

Following inclusion of this additional information, I would recommend the manuscript for publication in Nature Communications.

Response:

We agree with the referee that the LDOS measured with a metallic tip presents intensity for different amino acids, i.e. if measured at different locations along the sequence as shown in Figs 1e and S4. However, the differences are not large and hence do not lead to clear contrast between the arginine or tryptophan parts in the peptide sequences when imaged in dI/dV maps. Instead, dI/dV mapping of the peptide with a metallic tips results in an image presenting features for each amino acid, all at similar intensity, which does not merit a clear identification of any amino acid within the sequence.

Only the sensitised probe allows for an unambiguous assignment of the specific amino acid tryptophan in the LDOS map because by the rearrangement of R* at the tip apex enhances tunneling for tryptophane while suppressing intensity for others.

In order to clarify these points in the manuscript we have added the following text marked in blue.

We discuss the dI/dV spectra in Fig 1c in more detail, highlighting the observation of enhancement and suppression of the signal, which is leveraged to obtain clear contrast in dI/dV imaging.

The last four paragraphs of the subsection “**Molecular sensitisation of the probe for amino-acid identification**” now reads:

“With an unmodified metal tip, a weak and broad shoulder appears in the tunnelling spectra below -1.5 V if the tip is placed at the outer edges of the peptide dimer (marked in Fig. 1d). Likewise, in the centre of each molecule and at the peptide terminal within the dimer centre weak spectral feature can be observed in this voltage region.

The sensitised tip significantly alters the spectroscopic characteristic at the outer edges, *i.e.* a pronounced peak appears around -1.7 V in the differential conductance (Fig. 1c). *The dI/dV spectra measured in the centre of the peptide and at the peptide terminal within the dimer show reduced intensity at this voltage. Overall,* spectra taken at a larger energy interval and at all other locations on the molecule remain featureless, *suggesting they are nearly not affected* by sensitisation in contrast to the metal-probe spectrum (Supplementary Results – Sensitiser).

Using DFT calculations, the sensitisation enhanced spectral feature observed below -1.5 V can be assigned to the highest occupied molecular orbital (HOMO) of the molecule, which is localised in the WW section of the peptide (Supplementary Results – Molecular orbital). The spatial distribution of this enhanced feature can be mapped by tunnelling conductance maps measured at the corresponding energy (Fig. 1d). These conductance maps show distinct, intense features at the far sides of the dimer, *i.e.* at each outer end of the single peptide, whereas the middle part of the molecule and the inner dimer interface are featureless. *dI/dV maps recorded without tip functionalisation would show a large number of features of similar magnitude in which no one feature is distinct.*

Consequently, the R*-sensitised tip reveals the location of the tryptophan motif (WW) within the peptide due to the strong contrast in the dI/dV maps recorded at -1.6 V. Then the position of the other AA of the peptide can be identified in accord with theoretical calculations that optimise the molecular arrangement of the peptides on the surface (Supplementary Results – Structural model).”

The first paragraph of the **former Conclusion** section now reads:

“*In conclusion,* our results demonstrate the identification of the spatial position of the tryptophan amino acid W within peptide sequences. We observed an enhanced dI/dV signal at the local position of W using a R* sensitised probe that is generated by a reduction of the barrier and a concurrent conformational change at the sensitised probe. This enhancement is a *specific occurrence* and allows the allocation of W within a complex molecule like the peptides of 8-12 amino acids as used in our study and is thus a first step towards sequencing of peptides at surfaces. *A dI/dV image, in which many features in the same magnitude range would be visible without sensitisation, now presents singular features of much larger contrast. Because the structural change at the sensitised probe enhances some features while suppressing others, the discovered effect allows for chemically specific imaging.*”

Reviewer #2 (Remarks to the Author):

The authors have addressed all comments raised in the previous round of review.